# Effects of COVID-19 Pandemic on Students' Written Outcomes: An Interior Architecture Research/Theory Module Case Study in the UK

**Begüm Ulusoy** 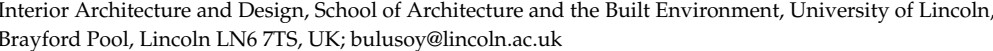

Interior Architecture and Design, School of Architecture and the Built Environment, University of Lincoln, Brayford Pool, Lincoln LN6 7TS, UK; bulusoy@lincoln.ac.uk

**Abstract:** Different learning methods (online, blended, blended-online and face-to-face) have been examined widely since the late 1990s. Although many design studies discuss engagement with these new methods in relation to studio modules, research/theory modules have not been investigated yet for interior architecture with both qualitative and quantitative data as a holistic approach. This study reveals how the new blended online learning method and the COVID-19 pandemic affected students' written outcomes in a research/theory module that accompanied their design module. For this purpose, the final written submissions of two year groups (2019–2020 vs. 2020–2021) are compared with both qualitative and quantitative analyses: their grades (performance), image (visual productivity) and reference (engagement with research) numbers and NVivo word count analyses (semantic analyses). The results show no significant difference between these two groups for both qualitative and quantitative analyses. Moreover, the study reveals that the numbers of images and references are good predictors for the grades of final-year students, thus showing their contribution to overall performance. Final-year research/theory modules in interior architecture might therefore be taught with blended online learning methods and can challenge, innovate and tailor studio teaching to contemporary needs. The study findings will be beneficial for educators and professionals, as well as managers, institution administrators, policymakers and decision-holders in HE who aim to employ blended online learning.

**Keywords:** interior architecture; blended online learning; COVID-19; research modules



## 1. Introduction

Until late 2019, 'pandemic' was not a familiar word for many people. Now, we are all affected and shaped by the COVID-19 outbreak (a.k.a. the pandemic) and its effects on our everyday lives: it changed us, our lives and our societies forever. There is no area that has not been affected by the pandemic and its consequences. While lockdowns and self-isolations were becoming normal parts of our lives, we learned to adapt ourselves to our residential interiors. We worked, studied, socialized and exercised in them and that changed how we experience interiors [1]. For design education, Marshalsey and Sclater [2] (p. 832) discussed the intersection of "physical and online environments with home/domestic environments". Higher education (HE) has been changing since early 2020 because the COVID-19 outbreak forced lecturers across the globe to convert their teaching strategies to fully online lessons while requiring their students to practice teaching and learning (T and L) activities in their homes and dormitories. Thus, our residential interiors quickly became our T and L interiors [3]. Almost all education institutions switched their traditional education systems online after the COVID-19 outbreak in March 2020. Their students, who previously had access to campus spaces, had to learn how to manage their learning in their residential interiors without any physical contact with lecturers, other students, a classroom and so on [3,4]. Meanwhile, lecturers struggled to adapt their teaching to the limitations of new and existing online platforms. Such facilities existed long before

the pandemic, but engagement with them was poor and uneven, meaning lecturers that had under-utilised such platforms struggled more than their more experienced colleagues. But the change had to happen within days for most educators and students. Without exception, students, parents and educators are affected by this shift. Yet, HE experienced some positive outcomes and experienced the advantages of online education e.g., instant feedback [5]. Marshalsey and Sclater [2] (p. 826) claimed that the "technological campuses of tomorrow have manifested" with the pandemic. Therefore, despite its tragically negative effects on education, the pandemic indirectly contributed to the future of T and L.

Most technologies frantically employed during the pandemic had been available for some time, but educators hesitated to use them in their teaching. Prensky [6] defined two groups: digital natives and digital immigrants. Digital natives refers to the generations that spent their formative years with technological innovations. On the other hand, digital immigrants were not born into a digital world and had to learn about this world later in their lives [6]. According to his study, these two groups are as different as people who are natives to a language and immigrants who have accents in those natives' language. Prensky [6] described digital immigrants as fascinated by new technologies despite only meeting them later in their lives. Yet, the last two decades proved that not all digital immigrants are fascinated by technology. On the contrary, some digital immigrants resist such technologies in their teaching, even if they are simultaneously using them for convenience in their daily lives. Mitra [7] states that the students of today are more familiar with and comfortable using online tools; therefore, online collaborative learning activities are more acceptable to them. Considering that online learning technologies will continue to advance, and contemporary students will become more skilful in using them, it can be predicted that using online learning tools will be a permanent and imperative part of HE in the future.

## 2. Literature Review

### 2.1. Face-to-Face (F2F) versus Online Design and Delivery (ODD)

There are several terms that might be used to define different online delivery methods. To avoid confusion, Power's [8] framework is embraced in this study (see Figure 1): online learning (OL), blending learning (BL) and blended online learning (BOL) as different learning methods for ODD (see Figure 1). Power [8] defined blended learning as both synchronous—a real-time interaction [7]—and asynchronous—offline activities that are sourced for an online course [7].

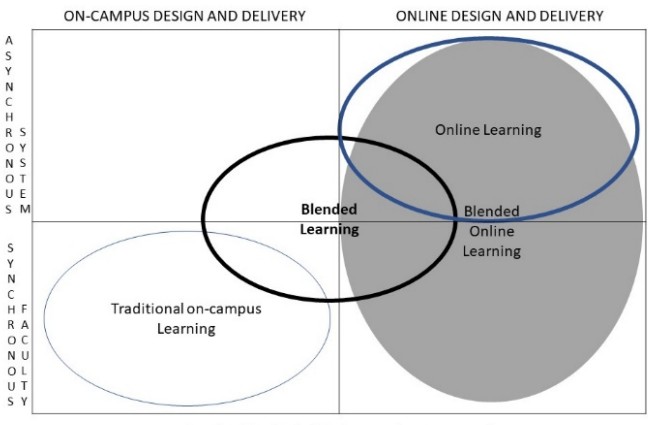

**Figure 1.** Blended online learning and its relationship to other learning methods (adapted from Power [8] (p. 510), which has a Creative Commons license).

Mitra [7] pointed out that online T and L is rooted in distance learning, a phenomenon with almost 300 years of history (see Figure 2). Distance learning has evolved with improvements to communication technologies, from radio to the internet [9], while it provides equal access to underrepresented groups (non-traditional students as mentioned by [10]. Simi-

larly, Sagun et al. [11] (p. 334) discussed how online T and L supports "disabled students who cannot physically attend in the classroom" and inclusiveness on a sociological level. Miller and Lu [10] reported many advantages of OL courses for non-traditional students before the pandemic (such as working students and people from lower economic classes) and their contribution to growing enrolment.

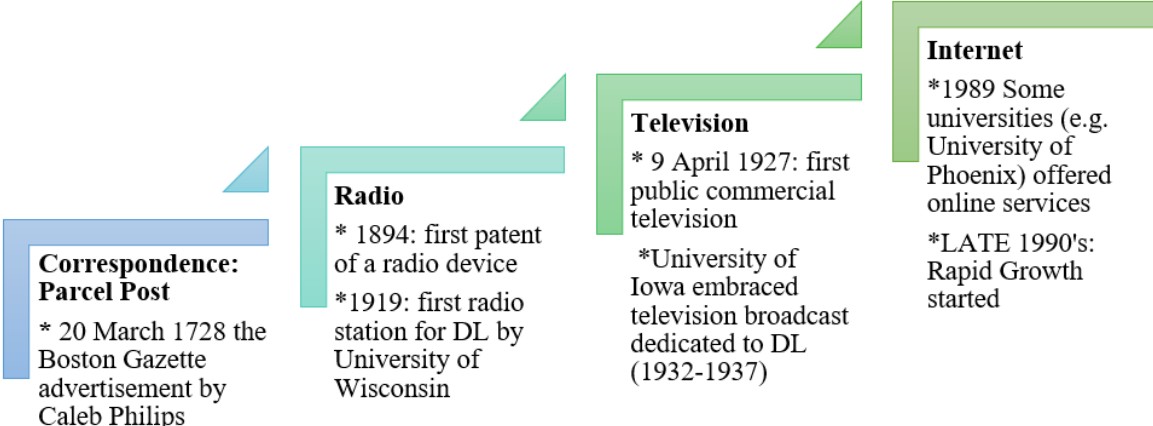

**Figure 2.** Evolution of Distance Learning (sourced from [9]).

One very visible and immersive change due to the pandemic was the migration from F2F to OL or BOL, and then (mostly after September 2020) for some institutions to BL. Within a relatively short time research studies emerged focused on its effects [12,13]. Lei and So [14] (p. 5) stated that even though online education is disadvantageous for students lacking discipline, its positive effects cannot be ignored. Online T and L requires fewer resources and staff and a smaller budget for HE institutions; thus, it is practically desirable [15]. Yakin and Linden [5] mentioned the negative effects (mostly technical problems and limited content) and positive effects (instant feedback and independent learning) of ODD on students and proposed that adaptive lessons enhance student engagement, motivation and performance. Online design and delivery provides flexibility (both time and space), opportunities for learners to repeat the same content as much as they need to and can benefit students even in hands-on courses such as dentistry [5].

An early study demonstrates that ODD has short-term (facilitating learning and improving curricula) and long-term (enhancing technology skills that increase employability) benefits for students [16]. Baker and Unni [17] (p. 46) revealed no differences in student satisfaction between online and F2F courses for both Asian and American students, which might be interpreted as a universal consideration of ODD. Moreover, they revealed the similar extended advantages of OL for traditional students, such as "enhanced communication among the learners" [17] (p. 50). Many studies mentioned the successful and smooth transition to ODD (with its learning methods: OL/BL/BOL) during the pandemic [5,12,18,19], with their success relying on teachers' engagement [20] and investment in technology and systems [12]. Marshalsey and Sclater [2], Dreamson [21] and Marshalsey [3] reported that previous experience with ODD provided benefits during the pandemic. Likewise, Park [22] suggested that teachers' capacity determined the success of OL courses, without taking into account the emergency transferring of design education to online formats during the pandemic. This proves the vital role of lecturers in the learning process.

Luckily, these novel learning methods were not totally new for HE. Some institutions had already transferred distance learning to OL and/or BOL after the internet had become an accessible option for their students [8]. A vast number of studies explore the difference between ODD and F2F learning, concluding that there is not any significant difference between the two (e.g., [23]). The website (No Significant Difference, available online: http://www.nosignificantdifference.org/ (accessed on 15 November 2021)) provides a number of sources that prove any differences between ODD and F2F methods are insignif-

icant. Although the website and the literature presented rigorous and consistent results, studies are relatively rare for the interior architecture and design (IAD) discipline before the pandemic [24]. There is no study comparing F2F and BOL methods with both qualitative and quantitative data as a holistic approach for research/theory module deliveries in IAD courses. One reason is that many IAD lecturers were not accustomed to ODD methods before the pandemic and their resistance meant the discipline could not adapt to changing technologies and skills for the interior architects/designers of the 21st century. Their unwillingness to embrace online elements in F2F teaching created a more challenging migration to ODD during the unpredictable and emergency conditions of the pandemic. From this perspective, the pandemic might be a blessing for those courses whose staff has reservations about ODD methods. More research studies can contribute to resolving their qualms.

### 2.2. Online Design and Delivery in Design and Architecture Education

A growing trend in education has seen more students enrolling in online courses every year [9]; however, only a "few fully online design courses" were available before 2020 [24] (p. 2). For example, The Open University has offered design courses for decades, the planning of which took years, which shows that the challenge design educators faced during the pandemic was overwhelming [25]. The pandemic accelerated the spread of online education across the globe, especially in disciplines such as design/architecture that were not taught mostly online [26]. Until the pandemic, IAD courses had relied on F2F methods, as was the case with many other design/architecture disciplines. Nevertheless, a substantial number of research studies before the pandemic explored and discussed online design studio education in several countries [11,15,22,27,28]. For example, Sagun et al. [11] discussed web-based education in an IAD course in Ankara, Turkey, more than two decades ago and one of the first MOOCs (massive open online course) in product design was launched in Delft, the Netherlands (Delft Design Approach MOOC), just a few years before the pandemic [15]. As Marshalsey and Sclater explain, "studio education is considered a signature pedagogy and has a distinct set of guiding principles such as facilitating critical play, thinking and making, and a pedagogy of ambiguity" [2] (p. 826). Dreamson [21] (p. 495) challenged the execution of traditional F2F studio teaching:

> *"In essence, the atelier model is often romanticised for design studios, yet its apprenticeship system could not be a sound approach in the digitally networked world where the speed of updating knowledge and skills through the network is tremendously faster than the transition from masters to apprentices . . . This means that design studios could no longer be the mainstream route for career development."*

One prominent reason for design education's lack of engagement with ODD was the lecturers' reluctance to deal with the challenges of these methods [10,21,26,27,29] before the pandemic, which forced them to improve and update their teaching skills under unprecedented conditions. Dreamson [21] (p. 485) stated that George [30] revealed critical barriers, one of which was "instructors' beliefs—studio-based learning cannot be replicated" and Dreamson [21] concluded that ODD's barriers and challenges are social components. Considering previous studies (e.g., [20,22]), lecturers and their commitment are very important for T and L and students, and their resistance and reluctance cost HE (socially and financially) and, potentially, come at the cost of student employability.

#### 2.2.1. Online Design and Delivery in the Design Studio

The studio is the main part of design education [22], and IAD learning, as with other design/architecture disciplines, consists of studio and lecture sessions [11]. The BOL and BL methods in the design studio have been embraced by interior architecture [11,27,28] and architecture disciplines [31]. One study before the pandemic [11] revealed the social, ideological, epistemological and pedagogical advantages of combining asynchronous and synchronous learning methods, which apply to BOL. Several research studies about online design education [2,3,11,15,22] reported positive effects from flexibility, accessibility,

recorded sessions, low budget, in-depth engagement, less distraction, avoiding everyday life necessities (e.g., commutes), better individual/group communication, personalisation, less formal communication with peers and lecturers and the easing of formality. After only two weeks, interior design studio students reported these advantages of online collaboration (most liked): ease of use/access; ease of sharing info and comments; convenience; the organization of materials in one place; and ease of reference [28] (pp. 483–484). Fleischmann [24] (p. 4) listed the further advantages of the online design studio, such as receiving feedback from outside experts, while claiming that there was no 'one size fits all' model. On the other hand, Jones and Lotz [4] (p. 4) mentioned several disadvantages and limitations of ODD such as the lack of "informal breakout spaces, etc.", which are hard to replace, alongside advantages such as international collaborations, making space for more voices than a traditional design studio and so on.

Sagun et al. [11] explained that online tools provide more control for students compared to physical studios, and shift students' engagement as they move from being passive listeners to more active learners. Ismail et al. [32] stated that the digital studio encourages dynamic and complex ideas. Iranmanesh and Onur [26] showed that the VDS (Virtual Design Studio) is superior to the PDS (Physical Design Studio) as it promotes self-dependence, a research-oriented approach, and provides more control for students (which underpins a student-oriented learning process). However, they did not propose the VDS as an alternative to the PDS, instead stating that a hybrid of the two with virtual reality might be the future of the design studio. Amro [20] and Alawad [29] explored the online interior architecture studio experience, Amro [20] stating that, although the pandemic caused a loss of motivation and high anxiety for design students whose online T and L required different approaches than other disciplines, that was overcome by their teachers' empathy, and students reported a positive experience in their overall design studio modules. Alawad [29] claimed that online the design studio is an attractive option that could enable the creative processes, and proposed combining the F2F and ODD methods' best properties, as do Pektaş [27] and Iranmanesh and Onur [26].

2.2.2. Online Design and Delivery in Research/Theory Modules

The BOL and BL methods in online research/theory modules have not been investigated as much as design studio modules in design education. Urban design [18] and fashion [33] disciplines reported positive changes in their T and L for research/theory modules with ODD. Peimani and Kamalipour [18] analysed their delivery before and after migration to online platforms due to the pandemic and the effects of BOL on T and L and concluded that challenging a fixed pedagogical framework is important for HE. Fernandes [33] focused on Millennials and GenZ—digital natives as defined by Prensky [6]—and aimed to integrate online group work as an innovative and productive assignment for a theoretical fashion module where they recorded positive effects on students. Online Design and Delivery offers many benefits, and previous studies show its positive effects on design/architecture disciplines. Interestingly, Marshalsey and Sclater [2] (p. 832) reported something that may appeal to lecturers who teach research/theory modules in IAD: "the student participants observed that online education had allowed them to study topics in more depth and detail, and that theoretical work was easier to comprehend".

Pektaş [27] proposed the blended studio environment, which sits well with the new generations' needs (a.k.a. digital natives) in design education and corresponds to social constructivist learning theories. They write that "social constructivist learning refers to an educational process that enables groups to create knowledge and meaning through co-creation" [27] (p. 694); thus, students become more active and independent participants in their own learning, as suggested by Sagun et al. [11] and Iranmanesh and Onur [26]. This also corresponds to Kolb's [34] experimental learning theory. A prior experimental interior architecture study [35], based on students' performance, revealed that all the learning styles from Kolb's theory [34] occur in design education and underscored that, through different design stages, all learning styles can be supported. Moreover, Zapalska and Brozik [36]

stated that all learning styles can be applied to online environments as well. Daalhuizen and Schoormans [15] discussed Kolb [34] as a prerequisite for design education, proving it could be successfully applied to fully online courses with dedicated didactic tools for reflective online teaching. As with these previous studies, Kolb's [34] experimental learning theory and social constructivist learning theory provide the theoretical framework for this paper. There remains a gap in the literature over how learning methods affect students' written work in IAD research/theory modules, yet the pandemic enabled a comparison of the written work of the 2019–2020 and 2020–2021 year groups. Dreamson [21] points out that there is a need to engage pedagogically now the peak of the pandemic has passed. This study aims to contribute to this pedagogical engagement by exploring the effects of the pandemic and BOL on student written outcomes. For this purpose, the BOL and F2F methods were compared through students' grades, the number of images and the number of references used in their submissions and semantic analyses of their final year written work in an IAD course. The following research question was asked in this study:

Research Question: How did the conversion of T and L from F2F to BOL during the pandemic affect overall student performance and their written work in a research/theory module within an IAD course?

Sub-questions:

1.  How were the semantic aspects of students' written work affected in the final year research/theory module for an IAD course, as a consequence of the pandemic?
2.  How was students' visual productivity affected during the final year research/theory module for an IAD course, as a consequence of the pandemic?
3.  How was students' engagement with research affected during the final year research/theory module for an IAD course, as a consequence of the pandemic?
4.  How were students' final grades affected during the final year research/theory module for an IAD course, as a consequence of the pandemic?
5.  How were students' grades and their number of visuals and references related to each other as an indicator of overall performance?

## 3. Methodology

This study aims to explore how the move to ODD enforced by the pandemic affected the written outcomes of student work using a single research/theory IAD module as a case study. Case studies, as a research method, have been employed by previous research studies from the entry-level [25] to the postgraduate level [37] and are proved to be a successful research method for design education studies in HE. Qualitative and quantitative methods were employed together to compare final year IAD final individual submissions for two different year groups (2019–2020, also known as 2020-year group: 30 students and 2020–2021, also known as 2021-year group: 19 students). All student works were examined, unless they submitted their work in the summer term as EC (extenuating circumstances) students.

### 3.1. The Research/Theory Module

Accompanying design modules, the aforementioned research module is taught throughout the academic year (see Figure 3), and it succeeds research modules of the first and second years (the course's website: https://www.lincoln.ac.uk/course/intintub/ (accessed on 12 December 2022)). It includes theoretical knowledge alongside research skills, through which students are asked to underpin their design process/outcomes with systematic research, and end with a submission of an academic research study as a written document with rich visuals corresponding to the IAD discipline (the course's website: https://www.lincoln.ac.uk/course/intintub/).

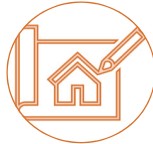

**Design Modules (Studio)**

**In IAD discipline:** the main part of design education (Park, 2011).

"...a shared place in which students are given practical tasks and projects to solve either individually or collaboratively and where students share their solutions or development processes with other students. Students and teacher in this space interact with each other based on traditional principles of supervision, consultation and discussion" (Park, 2011: 177).

**In the course of this case study,** three design modules are delivered throughout the academic year.

"The design process incorporates conceptual, technical, and professional knowledge areas, as well as conception development, resolution, and communication" (https://www.lincoln.ac.uk/course/intintub/).

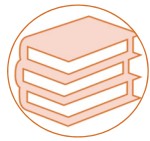

**Research/Theory Modules**

**In IAD discipline:** research/theory modules intent to serve studio modules (Ibrahim & Utaberta, 2012).

"Diverse subjects other than Design Studio offered in any architecture courses reflect the complexities integral in architecture" (Ibrahim & Utaberta, 2012: 30).

**In the course of this case study,** one research module run throughout the academic year.

"The research process stream focuses on design theory and contextual material, enabling students to develop research skills" (https://www.lincoln.ac.uk/course/intintub/).

**Figure 3.** Relation of research and design modules (the course's website: https://www.lincoln.ac.uk/course/intintub/).

Both year groups used Blackboard (a digital learning platform) for auxiliary services (lecture PowerPoint presentations, etc.) before the pandemic, as many other institutions had [13]. In addition to asynchronous sources on Blackboard, the 2021-year group started to use Microsoft Teams, which provides synchronous sessions (lectures/seminars/group and one-to-one tutorials) and its chat option for group and one-to-one confidential communication with their lecturers and peers. Synchronous sessions were recorded (except tutorials), and students were able to re-read chat communications, which minimized miscommunication and maximized students' access to content. Peimani and Kamalipour [18] state that Microsoft Teams is very user-friendly and supports lectures and seminar sessions and improves communication with reticent students. Its chat option contributes to further discussion before, during and after the sessions and improves the engagement of reticent students. Marshalsey and Sclater [2] revealed that students engage with chat boxes more than vocally (via the 'raising hand' option), which requires turning on their microphones. The chat option on Microsoft Teams, therefore, enables more student participation and benefits reticent students, as proposed by the literature [2,18] and observed in this module. Students were systematically taught and encouraged to have peer review, a fundamental skill for designers [18], from their first year on the course through their successive research modules. Daalhuizen and Schoormans [15] stated that receiving feedback and seeing other work provides an insightful opportunity for students to reflect and observe during their experimental learning process in a MOOC. For the 2021-year group, online meetings were recommended to students for peer review, which is a skill for the industry now.

The university enforces blind double-marking procedures, with two lecturers independently grading until finalizing their marks with the inclusion of a third lecturer if needed, ensuring the fair and objective assessment of student work. Both year groups in this study followed the same grading process; thus, any possible unconscious bias of the researcher, who is also the module leader, was avoided. Lei and So [14] stated that lecturers' teaching styles had prominent effects on student satisfaction and ensuring constant communication

was essential. In this study, the 2021-year group had regular access to their lecturers via emails, team meetings and online Q&A sessions. They were encouraged to communicate and raise their questions instantly to ensure their learning was not interrupted by the pandemic. Moreover, the university library provided uninterrupted support from the first lockdown in March 2020 and provided asynchronous sources that were already available (e.g., Harvard referencing handbooks), similarly to student support provided in previous studies [18]. Furthermore, the subject librarian had an academic writing session, and their team was available for both year groups.

*3.2. Data Collection/Analyses*

Both year groups submitted their documents on similar dates during their respective academic years (5 April 2020 and 30 March 2021). For both groups, only texts in students' works were analysed against their grades, number of images and number of references. Visual analyses (colour, content, etc.) of images were excluded since some of them were created for design modules that were in BL modules outside of lockdown periods. The study used data collected during the normal course of university business (grades, student work, etc.) to inform its findings. The decision to conduct this study began after teaching commenced, and data analysis began after the course ended. Students were therefore informed about the study after the completion of their course via an email with a brief summary of results. Because the study drew only on students' grades and the final outcomes of the module, and not experimentation with teaching delivery, their T and L was not affected by the study. However, it was shaped, changed and affected by the pandemic. Full ethical approval for this work with the Ethics Reference UoLReview Reference 2021_4026 (University of Lincoln) was received before the study began.

The data were analysed with both qualitative and quantitative approaches [38]. In the qualitative approach, the final outcomes of both groups were analysed and compared for both generalization and exact match results. The exact match provided some specific words such as 'pandemic' that were expected to be raised with the COVID-19 outbreak, whereas generalisations revealed concepts (see Figure 4). For quantitative analyses in both year groups, the first 1000 terms were compared through paired samples *t*-test in SPSS, which was used to reveal differences between the two groups. In generalisation word counts, some words were interchangeable unless they affected meaning (such as singulars-plurals, e.g., user vs. users). Moreover, grades and the number of images and references were analysed in order to reveal any significant relationship between them that revealed a holistic approach with the qualitative and quantitative data [39] of semantics (NVivo results with *t*-test for word counts) (see Figure 4).

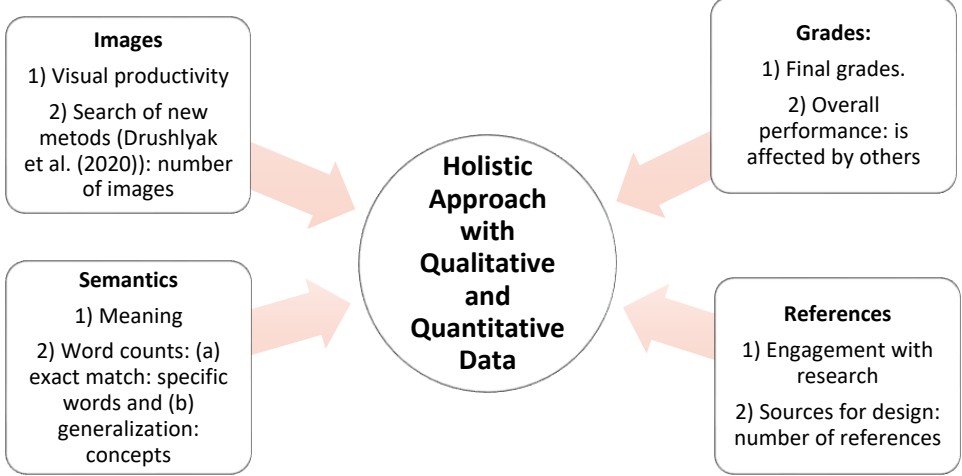

**Figure 4.** Analyses of students' written outcomes as a holistic approach.

### 3.3. Holistic Approach

For the research question (with sub-questions 1, 2, 3 and 4), students' written work was analysed in order to reveal its semantic aspects (semantics: "the study of meanings" according to the Merriam-Webster Dictionary, Semantics, available online: https://www.merriam-webster.com/dictionary/semantics (accessed on 12 December 2022)), their word counts, and a comparison of word counts between the two year groups alongside their grades and the number of images and references. In order to provide a neutral comparison, all student work was analysed through NVivo 12 as a reliable and objective qualitative data analysis software [40]. Word count analyses were used to understand "differences among participants" and were employed for at least three reasons: "(a) to identify patterns more easily, (b) to verify a hypothesis and (c) to maintain analytic integrity" [40] (p. 76). Each data set was grouped by 'word frequency' according to its semantic relationships, revealing word groups that provided context. The number of words and the number of images and references, compared against final grades, provided quantitative data. IBM SPSS Statistics 25 was used to analyse and compare word count outcomes (paired *t*-test) across the two groups.

Moreover, the research question (with sub-question 5) examined the relationship between performance, visual productivity and engagement with research within the research/theory module. For that purpose, students' final grades and the number of images (tables, drawings, etc.) and references were analysed, first with Pearson correlation and then Multiple Linear Regression. Final grades represent overall performance (i.e., [41]), which might be affected by images and references in the written work of students. Visual productivity "is associated with the search for new methods of solving problems" [42] (p. 151) and emerges from the human brain [43] (p. 251). Since in the scope of this study other aspects of images (forms, colours, etc.) cannot be analysed, the number of images were analysed to investigate visual productivity. Engagement with research is an essential part of the design process and an inseparable element of good design practice [44]; in this module, students' grades are inherently related to their research activities; therefore, a connection between engagement with research and their overall performance was investigated through the number of references as quantitative data. All these elements provide a holistic approach to students' outcomes and explore the design/research process for IAD with performance, visual productivity and engagement with research, and they underpin the comparison of the 2020 and 2021 year groups' semantic analyses (see Figure 4). This holistic approach, which embraces the triangulation of the data, ensures credibility in architectural research studies [44,45].

It should be noted that there are other variable factors aside from the pandemic, such as students' personalities, slight staff changes, etc. Nevertheless, because both student groups had the same brief and support from their lecturers, their submissions provide good data to investigate how migration to ODD affected their work. The 2021-year group had full BOL for the same module and were affected by the pandemic (i.e., no access to printed sources) and its other consequences. For example, the literature review shows that the pandemic affected the mental health of students with similar demographics [46] and they experienced anxiety [20]. This significant difference (moving to BOL due to the pandemic) between the two year groups overwhelms other contaminating effects; thus, students' final submissions are comparable in order to reveal the effects of the pandemic (and its consequences: ODD) on T and L. The brief, its criteria, learning outcomes and the content of the calendars of both years were the same; some major changes in design and delivery are revealed in Figure A1. Given that this module was not changed except for regular updates in lectures, presentations, etc., and converting its content to a BOL method, the 2020-year group functioned as a control group for this comparative study [7].

It was hypothesised that: (1) student written work would be affected due to ODD as a consequence of the pandemic and the two different year groups would differ in their outcomes; for example, semantic analyses would reveal statistically significant difference (quantitative data) and different concepts (qualitative data) in word counts; and (2) the

number of images and number of references used in assessments would have a positive correlation with grades.

## 4. Results and Discussion

### 4.1. Qualitative Analyses

Table 1 presents the only eight concepts that did not appear in the first one hundred terms for one of the groups after the first word count analyses. Figures and words for numbers are excluded, such as 2019, five, etc., for generalisation analysis, whereas only figures are excluded, such as twenty-five, for exact analysis. Moreover, the first 1000 terms of the initial analyses were word counted again by a generalisation feature. Table 2 presents the first 20 terms in which only 'cerebral' is not matched to the rest of the list. Word clouds were created after these analyses, presenting common concepts (see Figure 5). These results show that there are strong design-related elements that students embraced: artefacts, activities, content, etc., which were affected by neither the change in learning methods, nor the pandemic.

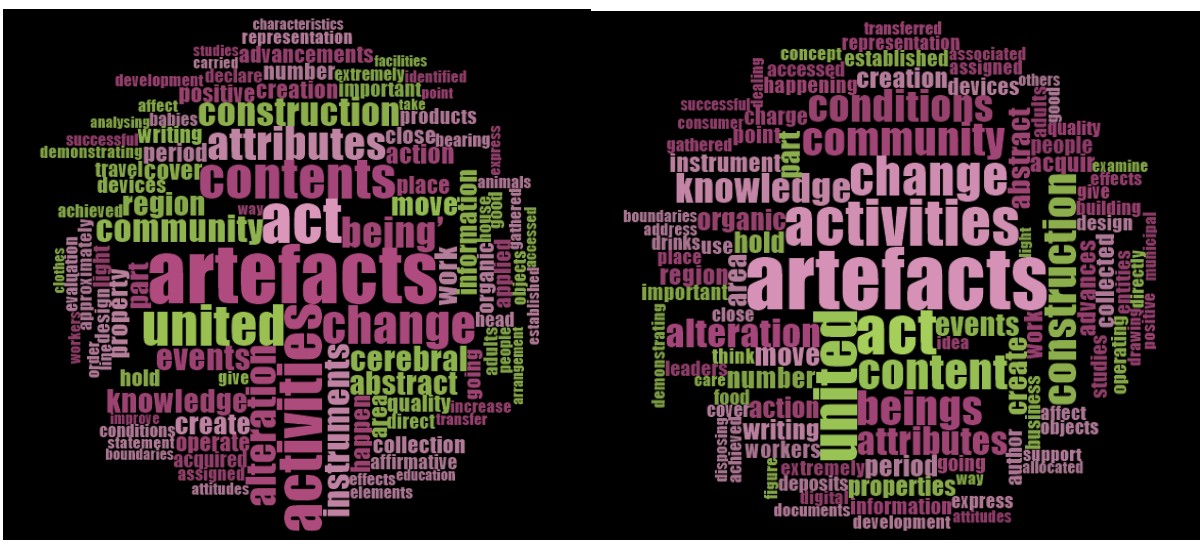

**Figure 5.** 2020 (**left**) vs. 2021 (**right**) word clouds of NVivo 12 analyses.

**Table 1.** Terms not revealed in the first word count analyses with generalisation function of NVivo 12.

| Word (Absent) | During the Pandemic (%) | Before the Pandemic (%) |
| --- | --- | --- |
| think | 0.23 | - |
| municipal | 0.19 | - |
| organisation | 0.14 | - |
| cerebral | - | 0.25 |
| collection | - | 0.16 |
| status | - | 0.13 |
| ethical | - | 0.11 |
| utilise | - | 0.10 |

**Table 2.** First 20 words after the second word count analyses of NVivo 12 (ranking order).

| Word 2020 | Word 2021 | Word 2020 | Word 2021 |
|---|---|---|---|
| artefacts | artefacts | community | knowledge |
| act | act | events | alteration |
| activities | activities | instruments | attributes |
| contents | united | cerebral | events |
| united | change | knowledge | abstract |
| change | content | abstract | create |
| attributes | beings | region | move |
| being | community | move | area |
| alteration | construction | work | hold |
| construction | conditions | create | number |

*4.2. Quantitative Analyses*

4.2.1. Word Counts

Word count analyses for the first 1000 words (based on weighted percentage) of both groups were compared through paired *t*-tests that revealed a comparison of their distribution based on their weightings. Only the matched terms of the initial analyses of both groups were used (see Table 3).

**Table 3.** Results of paired *t*-tests of word counts.

| Exact words (714 terms) | 2020-year group (M = 0.06 (2 dp), SD = 0.08) 2021-year group (M = 0.06 (2 dp), SD = 0.08 | No significant difference 0.0020, 95%CI [0.0040, 0.0000], t(712) = (1.92), $p = 0.055$ |
|---|---|---|
| Generalisation (708 terms) | 2020-year group (M = 0.05 (2 dp), SD = 0.06) 2021-year group (M = 0.05 (2 dp), SD = 0.06) | No significant difference 0.0000, 95%CI [0.0021, 0.0020], t(706) = (0.07), $p = 0.947$. |

The paired *t*-test reveals that there is no significant difference between year groups on word count results (exact and generalisations independently), although there are new vocabularies, such as pandemic (999th with 0.02 per cent) and COVID (808th with 0.02 per cent) in the 2021-year group's work. It can be ostensibly stated that BOL and F2F reveal similar outcomes in terms of research/theory modules in IAD and their written work semantically. The word lists were mostly dominated by design terms, common terms and pragmatic words (see Table 2 and Figure 5). It is important to note that these results showed no significant difference between the module's outcomes of the 2020 and 2021-year groups, although the pandemic affected students' perception, mental health and lifestyles as the literature suggested [20,46,47]. Savage et al. [46] stated that students' mental health was affected by the pandemic and their perceived stress increased during the first five weeks of the lockdown, with 214 university student participants examined whose demographics were very similar to this study's demographics (East Midlands, UK, mean age: 20, female percentage: 72) (see Table 4).

4.2.2. Grades versus Images or References

In order to explore student outcomes, their grades and the number of images and number of references were analysed separately (see Table 5). It is fruitful to note that the number of images and grades are not as different as the number of references, and the 2021-year group employed more sources than the 2020-year group. That might be an effect of the pandemic, during which students could not visit the library and did not have physical site visits, and they therefore tried to compensate by exploring more sources. For both groups, the Pearson correlation showed that grades and the number of images, and grades and the number of references, were significantly related (see Table 5).

**Table 4.** Demographics for both year groups.

| | 2020 | 2021 |
|---|---|---|
| Gender | Male: 20%<br>Female: 80% | Male: 16%<br>Female: 84% |
| Mean age (at the time of this study) | 22.06 | 22.26 |
| Nationality | UK: 73%<br>Malaysia: 17%<br>Syria: 3.3%<br>India: 3.3%<br>Zimbabwe: 3.3% | UK: 79%<br>Saudi Arabia: 5.2<br>Poland: 15.8 |

**Table 5.** Results of grades, number of visuals, number of references.

| | Average | Pearson Correlation | Multiple Regression with Enter Method | | |
|---|---|---|---|---|---|
| 2020-year group | Final grades: 68.03<br>Number of images: 59.86<br>Number of references: 40.13 | Significantly related | The model explained a statistically significant amount of variance in grades | $F_{(2, 27)} = 7.12$,<br>$p = 0.003$,<br>$R2 = 0.35$,<br>$R2adjusted = 0.30$. | An increase in one image corresponded, on average, to an increase grade 0.15 points, $B = 0.15$, $SD = 0.08$.<br>For each reference, a grade increased 0.25 points, $B = 0.25$, $SD = 0.11$. |
| | | $r = 0.47$,<br>$p = 0.008$,<br>$N = 30$ (number of images) | Number of images is a significant predictor of grades | $\beta = 0.34$,<br>$t_{(27)} = 2.02$,<br>$p = 0.053$. | |
| | | $r = 0.50$,<br>$p = 0.005$,<br>$N = 30$ (number of references) | Number of references also significantly predicted grades | $\beta = .37$,<br>$t_{(27)} = 2.24$,<br>$p = 0.034$. | |
| 2021-year group | Final grades: 59.94<br>Number of images: 60.94<br>Number of references: 54.94 | Significantly related | The model explained a statistically significant amount of variance in grades | $F_{(2, 16)} = 7.80$,<br>$p = 0.004$,<br>$R2 = 0.50$,<br>$R2adjusted = 0.43$. | An increase in one image corresponded, on average, to an increase in grade of 0.12 points, $B = 0.12$, $SD = 0.14$.<br>For each reference, a grade increased 0.28 points, $B = 0.28$, $SD = 0.17$. |
| | | $r = 0.63$,<br>$p = 0.003$,<br>$N = 19$ (number of images) | Number of images are a significant predictor of grades | $\beta = 0.25$,<br>$t_{(16)} = 0.85$,<br>$p = 0.41$. | |
| | | $r = 0.69$,<br>$p = 0.001$,<br>$N = 19$ (number of references) | Number of references also significantly predicted grades | $\beta = 0.50$,<br>$t_{(16)} = 1.67$,<br>$p = 0.12$. | |

Since correlation does not signify causation, further analyses were conducted to reveal their relationship. A multiple regression with the enter method was used to predict the grades of written documents from the number of their images and number of their references, separately for the two year groups. For the 2021-year group, the model is significant, which means the number of images and number of references can be used to predict grades, but they are not significant individually. Therefore, in order to predict grades, both the number of images and the number of references are required in the 2021-year group. For the 2020-year group, the model is significant, and both the number of images and the references are needed to predict a grade, although the number of references can be used as a good predicter independently in the 2020-year group unlike the 2021-year group (see Table 5). They are inherently related to overall performance as essential parts of this module. However, it is important to note that these causations between grades and the number of images or references are case specific and they should not be generalized for other cases (i.e., first year students). Further research is needed to uncover how visual productivity and engagement with research are related to overall performance in IAD.



It can be assumed that the pandemic was not ignored by students throughout this module. Yet, they were able to prioritize their work and control their perspectives towards the pandemic and its uncertainty, instead of letting this global disaster affect their learning fundamentally. The first hypothesis, which stated that the two groups would be significantly different, is rejected with these results. The second hypothesis, which suggested that grades are related to the number of images and number of references, cannot be rejected. This study's results are in line with the literature arguing that the teaching method does not affect students' performance [23,28] and that students "can learn in any type of environment and will gain new knowledge from their experiences regardless of the teaching modality" [48] (p. 6). Ergo, we should be focusing on "the assistance to learning aspect" instead of technology, as suggested by Larson and Sung [23] (p. 41). We need to embrace the positive effects of ODD, such as online discussions that mitigate student anxiety and increase their participation while encouraging critical thinking [16,23,26,28,48] and aim to mitigate the challenges and limitations of ODD such as the disadvantages of students lacking discipline [14] and technical problems and limited content [5]. Pektaş [27], Alawad [29] and Iranmanesh and Onur [26] discussed ODD should be embedded in the design studio, and research/theory modules, which were originally intended to serve studio learning [49], supposed to be following this approach. This study showed that students could accomplish similar results with the BOL method and its integration through their final year research/theory module(s) and its use in supporting IAD education might be a good practice during post-pandemic.

## 5. Conclusions

Distance learning, and its latest descendent, ODD, has long been offering advantages for both traditional and non-traditional students. Many researchers have argued that design disciplines including but not limited to IAD have fallen behind in embracing these new learning methods compared to other disciplines. Although the reasons can be discussed further, the contributions of distance learning to design learning are observable and prominent. Adding new activities online to existing courses improves the performance of students [5,7] and develops student engagement, motivation and perceived knowledge [5]. Many researchers (e.g., [20,27–29]) reported positive outcomes of online design studios, similar to Iranmanesh and Onur's [26] VDS. Nevertheless, previous studies showed converting the design studio is not a simple copy-paste task; there are several failed examples, and conversion requires the collaboration of all parties and rigorous hard work over several years [4]. Furthermore, in other design disciplines (e.g., fashion) researchers revealed good results for ODD methods in research/theory modules during the pandemic [18,33]. Online Design and Delivery comes with its shortcomings: Miller and Lu [10] point out the need for intentional and well-informed change in order to respect and protect intellectual knowledge, integrity and knowledge capacity and the management of HE while embedding ODD into F2F.

The pandemic forced educators to teach fully online (OL/BOL) in spring 2020 and then they all, voluntarily or involuntarily, migrated to these new methods (OL/BOL/BL) and had to intensively test ODD in design education.

*"In geography—which is all but ignored these days—there is no reason why a generation that can memorize over 100 Pokémon characters with all their characteristics, history and evolution can't learn the names, populations, capitals and relationships of all the 101 nations in the world. It just depends on how it is presented"* [6] (p. 6).

The generation that Prensky referred to in this quote are young professionals now after more than 20 years and the technological changes are mind-blowing compared to 2001. However, the rationale is still very relevant and important: the generation that can create the most inspiring videos on TikTok, Instagram, etc., should not be struggling to engage with the creative process of IAD because of methodology. Peimani and Kamalipour [18] (p. 4) stated that "the technology advocates echoed how the enforced online migration has contributed to the professionalisation of academics as pedagogues", which can bring them

to a better practice and digital transformation. One can argue that the digital transformation of design and architecture courses could have been achieved earlier, considering Prensky (and other researchers such as [11]) mentioned this in 2001 [6]. However, there was strong resistance from lecturers who are mostly digital immigrants (those who are not engaging with technology in their teaching), and their resistance could only be beaten by something as powerful as a pandemic. More recently, Dreamson [21] (p. 495) reacted to that resistance fairly:

> *"Rapid technological advancement has changed the landscape of education to be integrated with educational technology, and the worldwide pandemic has further accelerated its transition to digital learning and teaching. This process has not given educators and practitioners room for raising their resistant affection and making a pathetic excuse for not getting out of unfamiliarity and unawareness. Rather, they have been cast into the new learning environment."*

Despite resistance from some design and architecture lecturers, ODD in the design studio is possible. We must revisit the concept of the studio and how realistic it is to claim that current formats mimic the industry and its work environment. Teaching, in particular, has not changed since the Ecole Des Beaux Arts or Bauhaus [22], despite some visible adaptations from students, such as the use of laptops, and some technological developments, such as printers/3D printers in studios. As Dreamson [21] suggested, instructors may cling to a romantic idea of the atelier and lose the real purpose of design education: appropriately representing innovative industry applications (the design studio) with strong theoretical content (research/theory modules). Daalhuizen and Schoormans [15] showed that their dedicated learning tools had motivated students in fully online courses (OL in Figure 1) in terms of the experimental learning of Kolb [34]. Their tools included benchmark videos, sofa session videos, expert videos, peer reviews, which not only mimicked the physical design studio but engaged with the virtual nature of fully online teaching, which can be an inspirational step to the future of design education for IAD. For example, students experienced the positive effects of the benchmark (two master's students discussing and applying the same project as the students) and expert videos [15]. Challenging studio teaching inherently changes research/theory modules. Groat and Wang [44] (p. 21) stated that "the design and research constitute neither polar opposites nor equivalent domains of activity. Rather, the relationship between the two is far more nuanced, complementary, and robust". Design and research modules are closely related but are delivered differently, and in doing so advantage research modules in ODD, which contributes to the grasp of theoretical knowledge in depth and in detail [2]. This study's results show that research/theory modules might be taught fully online with BOL while innovating and tailoring IAD teaching to contemporary needs.

This study explored the effects and consequences of the pandemic on an IAD course, revealing the effects of ODD on final year students' work in the semantic content of their written submissions, their grades, the number of images included and the number of references. The study findings show no significant difference in the outcomes of the two year groups' work and the number of images and references included in their written submissions are good predictors of their final grades. However, it is important to note that both lecturers and students, working under the extraordinary conditions of the pandemic, might perform better if they were asked to migrate ODD under less exceptional circumstances. It is possible that both students and lecturers performed well because the extraordinary circumstances encouraged additional effort to compensate. Yet, HE has nevertheless changed significantly because of the pandemic and the new normal will require extra effort in areas of BOL. For the delivery of research/theory modules, BOL and F2F reveal similar results in terms of written student outcomes; thus, they could be converted to BOL permanently. However, future studies are needed for both design and research/theory modules in IAD, which reveal different perspectives from all parties: educators, students, decision-makers, policymakers, managers, etc. Moreover, the study did not investigate students' experiences through feedback and comments, which is a limitation of the method

that can be investigated in future studies in relation to students' written outcomes. In terms of the delivery of research/theory modules, future studies should focus on different year groups and their responses to ODD, and/or the combined effects of ODD on design and research modules in IAD. In doing so, more in-depth analyses can be conducted for visual productivity and engagement in relation to overall performance within IAD courses. The study findings will be beneficial for educators and professionals, as well as managers, institution administrators, policymakers and decision-holders in HE.

**Funding:** This research received no external funding.

**Institutional Review Board Statement:** The study was conducted in accordance with the Declaration of Helsinki, and approved by the Research Ethics Committee (REC) (Human Ethics Committee (PR)) of UNIVERSITY OF LINCOLN (UoLReview Reference 2021_4026 and 6 May 2021).

**Informed Consent Statement:** Informed consent was obtained from all subjects involved in the study via email. Because the study drew only on students' grades and the final outcomes of the module, and not experimentation with teaching delivery, their T and L was not affected by the study and they cannot be identified.

**Data Availability Statement:** The data presented in this study are available on request from the corresponding author. The data are not publicly available due to privacy and ethical restrictions.

**Acknowledgments:** I sincerely thank my late mother Serap Günay Ulusoy and my beloved father İsmail Orhan Ulusoy for their love, encouragement, and emotional, mental, and financial support. I'd like to thank Laura Pearson for their support during data analyses and Jon Coburn for his inspiring proofreading in addition to my students and colleagues at the University of Lincoln who have been with me during the pandemic. I'd like to thank the reviewers who contributed to this paper with their comments.

**Conflicts of Interest:** The author declares no conflict of interest.

## Appendix A. Major Changes Because of ODD

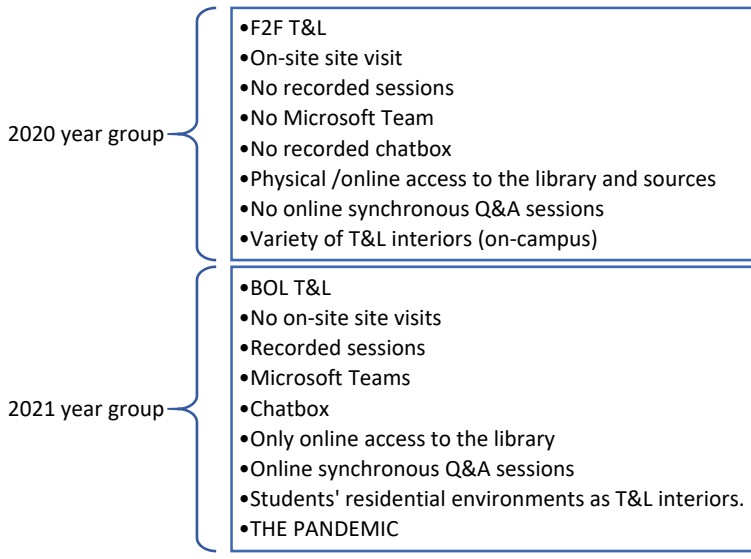

**Figure A1.** Major changes due to ODD enforced by the pandemic.

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
