# Peer review of "Effects of COVID-19 Pandemic on Students’ Written Outcomes: An Interior Architecture Research/Theory Module Case Study in the UK"

_education, doi:10.3390/educsci13010071_

Round 1
Reviewer 1 Report
The article explores the impacts of the Covid-19 pandemic on design education by drawing on a case study of an interior design architecture module in the context of the UK.
- The article needs to further clarify its aim and scope. While the title of the article suggests an exploration of the “effects of COVID-19 pandemic on design education”, it further outlines that it “reveals how the new blended online learning method and the COVID-19 pandemic affected students’ outcomes”. It seems that “design education” has been considered the same as “students’ outcomes”.
- The study seems to rely on rather reductionist assumptions. For example, the study seems to assume that the “number of images” indicates “visual performance” and/or the “number of references” indicates “in-depth research” (page 6). The following assumption seems unsubstantiated and quite problematic: “the number of images represent visual productivity; and the number of references stands for engagement with rigorous research process” (page 6). Including numerous images and references does not necessarily indicate critical engagement and/or in-depth research.
- The article could have provided more information regarding the specificities of the context such as the details of the related module design and delivery, its learning outcomes, its timeline of activities, its relations to the other modules, the related program of study, and the relevant institutional policy adaptations (before and during the COVID-19 pandemic in the UK).
- More relevant studies on learning and teaching experience, built environment education and/or design studio pedagogy (particularly in the context of the global pandemic in the UK) could have also been consulted in the article. For example, there are emerging research publications (mostly since 2020) on learning and teaching experience, built environment education and/or design studio pedagogy in relation to the global pandemic in the UK that could have been consulted as well.
- The research methodology requires significant improvement and further clarification to justify why and how certain decisions were made in the process of research design. For example, in what ways and to what extent can a comparison of the “students’ grades, the number of images and the number of references used in their submissions” reflect the differences/similarities between the BOL and F2F methods? and why is it important/necessary to analyse and compare “their semantic aspects, their word counts, and a comparison of word counts” as well as “their grades and number of images and references”?
- While it seems that an ethics approval was obtained before the study, it is quite concerning to see that “students were not informed throughout the study” (page 7). It is not clear how participants were informed and how an informed consent was obtained from participants for this study.
- The relevance of the outlined research question 2 (i.e., “How do students’ final grades, number of images, and number of references relate to one another in a research process module for an IAD course?”) to the aims and scope of this study is not clear.
- The use of the term “semantics” needs further clarification in the context of the relevant literature. Including both “word counts” and “meanings” as part of what is called “semantics” could benefit from clarification (figure 3, page 7). For example, please further elaborate on how “word counts” might be related to “meanings” and “semantics”.
- The relations and distinctions between “performance measure” and “overall performance” as well as between “number of references” and “engagement with rigorous in-depth research” need further clarification (figure 3, page 7). Please also further clarify the relations and distinctions between “number of images” and “image productivity” (figure 3, page 7).
- Please check the following references for accuracy: “Dreamson (2020: 485) and George (2017) underscored this barrier as "instructors’ beliefs – studio-based learning cannot be replicated" and concluded that OD education's barriers and challenges are social components” (page 4). It is not clear how the included direct quote might be related to both Dreamson (2020) and George (2017).
- The abstract section could have more clearly addressed the key contributions and implications of the study.
- The term “theory” is too broad as a listed keyword in the article.
- The article needs proofreading. There are several writing errors in the text (e.g., the abstract section).
Author Response
Reviewer-1:
The article explores the impacts of the Covid-19 pandemic on design education by drawing on a case study of an interior design architecture module in the context of the UK.
First of all, I would like to thank you for the time and effort put into the review of the paper. Your work on this article is much appreciated and helpful.
- The article needs to further clarify its aim and scope. While the title of the article suggests an exploration of the “effects of COVID-19 pandemic on design education”, it further outlines that it “reveals how the new blended online learning method and the COVID-19 pandemic affected students’ outcomes”. It seems that “design education” has been considered the same as “students’ outcomes”.
The title and the abstract were revised and students’ outcomes were clarified further, showing the aim and scope of the study clearly. That is followed throughout the paper during its revision. The title was revised to reveal the study focuses on students’ written outcomes. See Title and Abstract
- The study seems to rely on rather reductionist assumptions. For example, the study seems to assume that the “number of images” indicates “visual performance” and/or the “number of references” indicates “in-depth research” (page 6). The following assumption seems unsubstantiated and quite problematic: “the number of images represent visual productivity; and the number of references stands for engagement with rigorous research process” (page 6). Including numerous images and references does not necessarily indicate critical engagement and/or in-depth research.
Further discussion on visual productivity and engagement with research is added (lines 403-421) to improve clarity and to provide further explanations about these concepts. The aforementioned sentence is replaced with a more insightful discussion (page 9-10, 3.3. Holistic Approach and page 16, lines 597-602). These concepts are analysed since they are essential parts of the research modules in interior architecture and their contribution is fundamental for student performance. However, to the best of our knowledge, there is not any solid theory about them yet in interior architecture studies, therefore, after an in-depth literature review, we proposed their connection should be fruitful to discuss. The literature mostly employ the grades of students as a measurement of performance. For example, Pektas (2014)[1] used the term grades of students. We aim to analyse and explore other factors, which may contribute to overall performance in addition to grades, which would be fruitful. The revised paper reveals a discussion (lines 579-602) and suggests future studies (lines 739-741) about these topics. Further explanation of these terms can be found in points 5, 8, and 9 in this letter.
- The article could have provided more information regarding the specificities of the context such as the details of the related module design and delivery, its learning outcomes, its timeline of activities, its relations to the other modules, the related program of study, and the relevant institutional policy adaptations (before and during the COVID-19 pandemic in the UK).
This feedback is so helpful to change Appendix 1, revise the methodology section and add new figures. The table in Appendix 1, which revealed the criteria of the module, was replaced with a comparison figure (the comparison of major differences between two year groups) because the criteria (and learning outcomes, the content of the calendar, etc.) of the brief were not changed between two-year groups and the previous table did not contribute to the paper. Similar information about the module is already available on page 7 (lines 311-317). Moreover, the module is elaborated in 3.1. The Research/Theory Module (p.7). These decisions improved the paper significantly. Further details are added as an image to reveal the design-research modules’ relationship and their delivery (see page 8, Figure 3). Although we appreciate all recommendations we did not add all elements aforementioned above because they are excessively long documents, they were not changed between two year groups, and the University and other colleagues hold the copyright of them. However, we added the website of the course (Removed for peer review) (line 313) which reveals the aforementioned information further. If the article will be published, the website becomes available to the reader which will uncover more details about the course and its modules.
- More relevant studies on learning and teaching experience, built environment education and/or design studio pedagogy (particularly in the context of the global pandemic in the UK) could have also been consulted in the article. For example, there are emerging research publications (mostly since 2020) on learning and teaching experience, built environment education and/or design studio pedagogy in relation to the global pandemic in the UK that could have been consulted as well.
Thank you for your recommendations about sources, relevant sources are added to the article. We particularly focus on student outcomes in a research module of an interior architecture course. Therefore, your recommendation about a particular direction is very well landed in our paper (see Reference List).
- The research methodology requires significant improvement and further clarification to justify why and how certain decisions were made in the process of research design. For example, in what ways and to what extent can a comparison of the “students’ grades, the number of images and the number of references used in their submissions” reflect the differences/similarities between the BOL and F2F methods? and why is it important/necessary to analyse and compare “their semantic aspects, their word counts, and a comparison of word counts” as well as “their grades and number of images and references”?
The methodology section is revised significantly to clarify how these concepts are related to each other and the scope of the study. The connection between qualitative data and quantitative data of semantics is linked to underpin a holistic approach with grades, images, and references, to the effects of the pandemic and BOL on student written outcomes (lines 416-421). Further details are available in 3.3. Holistic Approach.
- While it seems that an ethics approval was obtained before the study, it is quite concerning to see that “students were not informed throughout the study” (page 7). It is not clear how participants were informed and how an informed consent was obtained from participants for this study.
The study used data collected during the normal course of university business (grades, student work etc.) to inform its findings. Teaching took place as normal throughout the academic year and the students received an email with a short summary of results to inform them about the research after the analyses were completed and they were asked to inform the researcher/module leader if they wanted their work to be excluded from these analyses. In doing so we avoid any bias, and the study has no effect on their learning process since the changes were enforced by the pandemic, not by a stimulus that is designed by the researchers. Moreover, students’ work, personal information, etc. are not identifiable in the study. If required, we can provide all emails to the students. This procedure is clarified in the revised paper (lines 364-372). Ethical approval is received with Ethical Reference (Removed for peer review), (University of Removed for peer review). If required, the letter can be provided.
- The relevance of the outlined research question 2 (i.e., “How do students’ final grades, number of images, and number of references relate to one another in a research process module for an IAD course?”) to the aims and scope of this study is not clear.
The research questions are revised to improve clarity, which was a great contribution to the paper. The question is written as one overarching question with sub-questions in order to show how data and the research method are employed following the aims and scope of the study (lines 274-294).
- The use of the term “semantics” needs further clarification in the context of the relevant literature. Including both “word counts” and “meanings” as part of what is called “semantics” could benefit from clarification (figure 3, page 7). For example, please further elaborate on how “word counts” might be related to “meanings” and “semantics”.
The figure is revised and redrawn to reveal more details about these decisions and text is added to the revised paper(page 11, Figure 4) for further clarification. The main aim is to uncover the effects of the pandemic and BOL on final-year student written outcomes in a research/theory module with a holistic approach (both qualitative and quantitative data). For semantics, Nvivo results (qualitative) and word counts – t-test (quantitative) data provide different aspects to analyse the written outcomes of students. (lines 389-392)
- The relations and distinctions between “performance measure” and “overall performance” as well as between “number of references” and “engagement with rigorous in-depth research” need further clarification (figure 3, page 7). Please also further clarify the relations and distinctions between “number of images” and “image productivity” (figure 3, page 7).
References, images, meaning in the text with concepts, and grades were employed to show how the pandemic might affect different elements in a research module in interior architecture as an overall performance. These terms are revised and clarified in the revised manuscript to improve clarity and further critical discussions are provided. Results about a number of images and references in relation to grades should not be generalised and should be discussed as a part of this case study (lines 579-602). Future research is needed to reveal their relationships in different circumstances (lines 739-741). Grades are used for overall performance before as many research studies mentioned and ‘visual productivity’ and ‘engagement with research’ with both qualitative and quantitative analyses provide a holistic approach (Figure 4). This holistic approach ensures triangulation[2],[3] (ensures credibility in architectural research studies) of the data instead of using only the final grades of students.
- Please check the following references for accuracy: “Dreamson (2020: 485) and George (2017) underscored this barrier as "instructors’ beliefs – studio-based learning cannot be replicated" and concluded that OD education's barriers and challenges are social components” (page 4). It is not clear how the included direct quote might be related to both Dreamson (2020) and George (2017).
The sentence is rewritten to improve its clarity (lines 183-186)
- The abstract section could have more clearly addressed the key contributions and implications of the study.
The abstract is revised to improve the clarity and key contributions and implications are added. Please see Abstract.
- The term “theory” is too broad as a listed keyword in the article.
This too-broad keyword is removed and replaced with research/theory modules. Please see Keywords.
- The article needs proofreading. There are several writing errors in the text (e.g., the abstract section).
The proofreading is completed by a native-English-speaking senior lecturer with a Ph.D. degree.
[1] PektaÅŸ, Åž. T. (2014). Correlations between the visualizer/imager cognitive style and achievement in digital modeling tasks. Procedia-Social and Behavioral Sciences, 116, 5053-5057.
[2] Martin, B. & Hanington, B. (2018). The Pocket Universal Methods of Design. Quarto Publishing Group USA Inc.
[3] Groat, L. N., & Wang, D. (2013). Architectural research methods. John Wiley & Sons.
Reviewer 2 Report
The impacts of COVID-19 pandemic and its associated uncertainties on design education continues to be a topic of much debate within built environment professions and design-related disciplines. As such, the current paper seeks to explore a relevant topic and can potentially be interesting to the readers of the education sciences journal. However, the paper currently poses several questions and concerns and I recommend that a major revision is warranted.
Please find below my concerns with further details. I ask that the author(s) must address the following comments in their response:
- The gap and its significance should be better explained in the abstract. "many studies discuss engagement with online learning..." Not clear if the author refers to the studies in the context of design education?
The findings (outlined in the abstract) do not seem to be able to show what has been added to the existing literature/scholarship.
- The flow of narrative in the introduction section could significantly benefit from clarity.
It is not clearly discernible what each paragraph is pointing to.
- It is not clear what “the oldest interior typology” means. The author only mentions houses. It is not entirely clear why typology is used in this context?
- When the author says “Almost all education institutions switched their traditional education systems 34 online” it is not clear which context and what year this refers to?
- “Students, used to enjoying campus space for both social events and learning before 35 the pandemic, had to learn how to manage their learning in their residential interiors 36 without any physical contact with lecturers, other students, or a classroom.” This lack supporting evidence.
- Almost Lacks critical engagement with the literature. For instance, if it says during Covid some studies point to mental health of students and some introduce adaptive approaches in design. It’s not clear what key argument from these studies are important here and how they are linked to the research questions in this paper. This is something that seems a recurring issue in most parts of the literature review.
What is the key focus in each paragraph of Literature Review is not entirely clear.
- “The web site http://www.nosignifi-cantdifference.org/ …” The in-text citation could benefit from more consistency.
- While the study endeavours to justify the gap in the IAD discipline in the last part of the section 1.1., there has been limited literature that explains the use of blended and/or face-to-face learning and teaching in other design disciplines. There has been recent published work in Education Sciences that might be of interest (e.g., future of design studio education, Learning and teaching urban design through design studio pedagogy and the like)
- The key focus in the first paragraph of the section 1.2. is not entirely clear; is it about the difference between post pandemic and pre-pandemic differences in the delivery and design of the modules in design-related disciplines? The section should be rewritten so that the key focus becomes clearer. Perhaps, one way to do this is to restructure your text according to certain clear questions and leave out those sentences which are not relevant.
- If the paragraph (starting with line 172) is looking at the use of blended approach in design-related disciplines/fields, it can also explore other new publications which address the use of BL and its capacities and challenges in relation to learning and teaching activities in other related fields such as urban design. Also, the paragraph is quite a long one focusing heavily on the advantages and positive aspects of the online learning almost without critical engagement with the challenges and limitations of this mode of delivery.
- Please make sure that the paper is proofread. (e.g., “… proposed blended the learning studio environment…” in line 215)
- In line 230, “for this case study”: I think this needs to be replaced by this paper as the methodology including the selected case study has not yet been introduced!
- In line 242, “research process module”: what does this mean? This is not consistent with other parts of the paper where the module has been called differently (e.g., research/theory IAD, research module). This needs to be clarified in the paper.
- Methodology: It would be helpful if the author used a table which included the specifics regarding the design and delivery of the research module for the 2019-2020 and 2020-2021 year groups. The research design decisions should also be explained and justified in the context of relevant literature (e.g., case study approach in the context of design education)
It will also make more sense if the author elaborates further on the selection of those criteria to analyse students' outcomes (e.g., word counts)
- Microsoft Teams in line 301: If the author has only engaged with the relevant literature to discuss the capacities of Microsoft Teams without relying on their students' comments and feedback, then this should be written as part of the limitations of the methods.
- “ … enables more student participation and benefits shy students.” In line 311: what is the evidence for this? Is this based on the observation of the module leader/instructor/?
- The first section of the results and discussion seems to be better placed in the methodology section.
- In the conclusion section (paragraph starting with line 488): The distinction between the design studio modules and research/theory modules should be better explained particularly given the associated capacities and challenges of ODD and BOL modes of delivery.
Author Response
Reviwer-2:
The impacts of COVID-19 pandemic and its associated uncertainties on design education continues to be a topic of much debate within built environment professions and design-related disciplines. As such, the current paper seeks to explore a relevant topic and can potentially be interesting to the readers of the education sciences journal. However, the paper currently poses several questions and concerns and I recommend that a major revision is warranted.
Please find below my concerns with further details. I ask that the author(s) must address the following comments in their response:
Thank you so much for your recommendation, very much appreciated. We address all points in the revised paper, please find our responses below:
- The gap and its significance should be better explained in the abstract. "many studies discuss engagement with online learning..." Not clear if the author refers to the studies in the context of design education? The findings (outlined in the abstract) do not seem to be able to show what has been added to the existing literature/scholarship.
The abstract is revised to increase clarity and engagement with readers and the findings are elaborated Please see the Abstract.
- The flow of narrative in the introduction section could significantly benefit from clarity. It is not clearly discernible what each paragraph is pointing to.
Sections in the paper are revised and further clarity is provided. We divided the long introduction section and added a “Literature Review” section with its subsections in order to present a clear flow of the narrative. Moreover, there are further revisions in paragraphs to improve clarity. Please see the Introduction and Literature Review.
- It is not clear what “the oldest interior typology” means. The author only mentions houses. It is not entirely clear why typology is used in this context?
Thank you for pointing out that, removing this irrelevant term contributes to the paper’s clarity (page 1 line 34).
- When the author says “Almost all education institutions switched their traditional education systems 34 online” it is not clear which context and what year this refers to?
Further clarity is provided and the year the pandemic started (2020) is added to the relevant paragraph (page 1, lines 42-43).
- “Students, used to enjoying campus space for both social events and learning before 35 the pandemic, had to learn how to manage their learning in their residential interiors 36 without any physical contact with lecturers, other students, or a classroom.” This lack supporting evidence.
The sentence is revised to improve clarity. It reveals the shift in student engagement from physical to fully online or blended modes due to the pandemic and the relevant references are added (pages 1-2, lines 44-47)
- Almost Lacks critical engagement with the literature. For instance, if it says during Covid some studies point to mental health of students and some introduce adaptive approaches in design. It’s not clear what key argument from these studies are important here and how they are linked to the research questions in this paper. This is something that seems a recurring issue in most parts of the literature review. What is the key focus in each paragraph of Literature Review is not entirely clear.
Paragraphs of the introduction and the literature review are revised and rearranged, and irrelevant sentences are removed to improve clarity (please see the Introduction and Literature Review). Further, more relevant sources are added to reveal the key focus and connect them to the existing sources. For critical engagement and in-depth research, we added more relevant sources, exclude some less relevant sources/sentences (Please see the Reference list), and include some different aspects from the existing literature about case studies in design education (lines 301-304) and the negative effects of ODD on students and HE (for examples, lines 207-211).
- “The web site http://www.nosignifi-cantdifference.org/ …” The in-text citation could benefit from more consistency.
This web link is moved to the reference list and its in-text citation is replaced with the website’s title (line 135).
- While the study endeavours to justify the gap in the IAD discipline in the last part of the section 1.1., there has been limited literature that explains the use of blended and/or face-to-face learning and teaching in other design disciplines. There has been recent published work in Education Sciences that might be of interest (e.g., future of design studio education, Learning and teaching urban design through design studio pedagogy and the like)
Thank you so much for this recommendation, further sources from Education Sciences are investigated and added to the manuscript (please see the Reference List).
- The key focus in the first paragraph of the section 1.2. is not entirely clear; is it about the difference between post pandemic and pre-pandemic differences in the delivery and design of the modules in design-related disciplines? The section should be rewritten so that the key focus becomes clearer. Perhaps, one way to do this is to restructure your text according to certain clear questions and leave out those sentences which are not relevant.
Research questions are revised (lines 274-294), and the paragraphs of the Introduction are restructured (with a new Literature Review section and its subsections) in order to address this problem and improve the clarity of the paper (please see the Introduction and Literature Review). Irrelevant sentences throughout the paper are removed, as recommended, to provide further clarity.
- If the paragraph (starting with line 172) is looking at the use of blended approach in design-related disciplines/fields, it can also explore other new publications which address the use of BL and its capacities and challenges in relation to learning and teaching activities in other related fields such as urban design. Also, the paragraph is quite a long one focusing heavily on the advantages and positive aspects of the online learning almost without critical engagement with the challenges and limitations of this mode of delivery.
The long paragraph is restructured as shorter paragraphs (pages 5-7) and further negative aspects of ODD are discussed (lines 207-211 in addition to lines 638-640 and lines 657-659) in the revised paper in order to reveal challenges and limitations of ODD. Further sources are added (please see the Reference List).
- Please make sure that the paper is proofread. (e.g., “… proposed blended the learning studio environment…” in line 215)
The proofreading is completed by a native-English-speaking senior lecturer with a Ph.D. degree.
- In line 230, “for this case study”: I think this needs to be replaced by this paper as the methodology including the selected case study has not yet been introduced!
“For this case study” is replaced with “for this paper” (line 264).
- In line 242, “research process module”: what does this mean? This is not consistent with other parts of the paper where the module has been called differently (e.g., research/theory IAD, research module). This needs to be clarified in the paper.
The “research process module” term, which is confusing, is removed (line 277). The research questions are revised (lines 274-294)
- Methodology: It would be helpful if the author used a table which included the specifics regarding the design and delivery of the research module for the 2019-2020 and 2020-2021 year groups. The research design decisions should also be explained and justified in the context of relevant literature (e.g., case study approach in the context of design education). It will also make more sense if the author elaborates further on the selection of those criteria to analyse students' outcomes (e.g., word counts)
Further information about the module is added to Appendix 1 with another visual to reveal the relationship between research and design modules (page 8, Figure 3). The research design decisions in relation to the case study method are explained (lines 301-304). Further discussion on selection criteria is provided (3.3. Holistic Approach, first two paragraphs).
- Microsoft Teams in line 301: If the author has only engaged with the relevant literature to discuss the capacities of Microsoft Teams without relying on their students' comments and feedback, then this should be written as part of the limitations of the methods.
We engaged with the relevant literature and reflected on our own observations. This is explained in the paper and elaborated as a limitation of the methods in the Conclusion section (lines 734-737).
- “ … enables more student participation and benefits shy students.” In line 311: what is the evidence for this? Is this based on the observation of the module leader/instructor/?
Reference about this statement is added, moreover, it is clarified that this was observed by the researcher/module leader throughout the year (lines 335-338).
- The first section of the results and discussion seems to be better placed in the methodology section.
The first section of the results and discussion is placed in the methodology which contributes to the flow of the narrative. We appreciate this comment very much! (page 9)
- In the conclusion section (paragraph starting with line 488): The distinction between the design studio modules and research/theory modules should be better explained particularly given the associated capacities and challenges of ODD and BOL modes of delivery.
An image is added to reveal these differences which improve the clarity of the paper in terms of design studio and research/theory modules. And the paragraph starting with line 694 (lines 711-718) is revised to provide further explanations.
Reviewer 3 Report
This is a well-written paper, which discusses a chronological timeline of distance, online and blended learning and which has a rigorous literature review addressing multiple aspects of the effects of the pandemic on T&L. You need to foreground in the title and/or the RQ that this study is about how the pandemic affected the written outcomes of students works.
The RQ's could perhaps be condensed into one overarching RQ, with supporting sub-question, as currently, they are confusing as a set of 2? E.g.,
Research Question 1: How did the conversion of T&L from F2F to BOL during the pandemic affect student performance and their written outcomes within an IAD course?
Potential sub-questions:
1: How were the semantic aspects an indicator of student performance, h in a research process module for an IAD course, as a consequence of the pandemic?
2: How was students' visual productivity affected during the research process for an IAD course, as a consequence of the pandemic?
And so on... separate the issues investigated.
Line 47 - missing full stop.
Line 162 - missing 'the' before 'lecturer' OR lecturers'
Line 319 has confusing language at the beginning - 'The university use for this study enforces blind double marking'
Please also look at the papers written by Derek Jones and others:
Jones, D 2022, ‘Studio Use in Distance Design Education’, ProQuest Dissertations Publishing.
Jones, D & Lotz, N 2021, ‘Design Education: Teaching in Crisis’, Design and Technology Education, vol. 26, no. 4, p. 4.
Jones, D 2014, ‘Reading students' minds : design assessment in distance education’, Journal of Learning Design, vol. 7, no. 1, pp. 27–39.
Author Response
Reviewer-3:
This is a well-written paper, which discusses a chronological timeline of distance, online and blended learning and which has a rigorous literature review addressing multiple aspects of the effects of the pandemic on T&L.
Thank you so much for your recommendation and insightful comments, very much appreciated. We address all points in the revised paper, please find our responses below:
- You need to foreground in the title and/or the RQ that this study is about how the pandemic affected the written outcomes of students works. The RQ's could perhaps be condensed into one overarching RQ, with supporting sub-question, as currently, they are confusing as a set of 2? E.g.,
Research Question 1: How did the conversion of T&L from F2F to BOL during the pandemic affect student performance and their written outcomes within an IAD course?
Potential sub-questions:
1: How were the semantic aspects an indicator of student performance, h in a research process module for an IAD course, as a consequence of the pandemic?
2: How was students' visual productivity affected during the research process for an IAD course, as a consequence of the pandemic?
And so on... separate the issues investigated.
This feedback is much appreciated. The research question (lines 274-294) and its new sub-questions are revised accordingly. The title was revised to reveal the study focuses on students’ written outcomes.
- Line 47 - missing full stop.
Line 162 - missing 'the' before 'lecturer' OR lecturers'
Line 319 has confusing language at the beginning - 'The university use for this study enforces blind double marking'
All the above-mentioned mistakes are fixed in the text.
- Please also look at the papers written by Derek Jones and others:
Jones, D 2022, ‘Studio Use in Distance Design Education’, ProQuest Dissertations Publishing.
Jones, D & Lotz, N 2021, ‘Design Education: Teaching in Crisis’, Design and Technology Education, vol. 26, no. 4, p. 4.
Jones, D 2014, ‘Reading students' minds : design assessment in distance education’, Journal of Learning Design, vol. 7, no. 1, pp. 27–39.
Many thanks for these recommendations that improve the paper. Recommended sources with other sources are added to the paper (please see the Reference List).
Reviewer 4 Report
During this post-pandemic era, the paper treats an essential topic in design education correspondingly and with which reveals the author(s)’ contributions. However, some minor revision is still necessary to enhance its academic value.
1. Author(s) included and conducted a thoroughgoing literature review in section 1. It could be more appropriate to separate the “Literature Review” from the “Introduction.”
2. Identifying the “design module” and “research module” in design education will be helpful; the illustrated framework or figure related to those modules will also bring a better clarification.
3. It would be conducive to comprehending the research findings by employing the simplified “Tables” to demonstrate the statistical information in section 3.2 “Quantitative Analyses.”
4. The supplement of teaching plans about “Interior Architecture” in contrast with two different years groups will facilitate understanding the changing teaching conversion from F2F to ODD. It could be complemented as Appendix material.
Author Response
Reviewer-4:
During this post-pandemic era, the paper treats an essential topic in design education correspondingly and with which reveals the author(s)’ contributions. However, some minor revision is still necessary to enhance its academic value.
Thank you so much for your recommendation and very helpful comments, very much appreciated. We address all points in the revised paper, please find our responses below:
- Author(s) included and conducted a thoroughgoing literature review in section 1. It could be more appropriate to separate the “Literature Review” from the “Introduction.”
Thank you so much for this recommendation! We revised the Introduction section by adding a new section “Literature Review” with its subsections which significantly improved the flow of narrative and clarity. (please see the Introduction).
- Identifying the “design module” and “research module” in design education will be helpful; the illustrated framework or figure related to those modules will also bring a better clarification.
We added a new image to reveal how design modules and research modules are related to each other in both our IAD course and the IAD discipline (page 8, Figure 3).
- It would be conducive to comprehending the research findings by employing the simplified “Tables” to demonstrate the statistical information in section 3.2 “Quantitative Analyses.”
Tables (page 14 Table 3 and pages 15-16 Table 5) are added in order to present a more engaging Results section.
- The supplement of teaching plans about “Interior Architecture” in contrast with two different years groups will facilitate understanding the changing teaching conversion from F2F to ODD. It could be complemented as Appendix material.
In order to reveal the most influential changes between F2F and ODD we added a new image as Appendix 1. We prefer not to add more information about the course/module because they are excessively long documents and they were not changed between two year groups, and the University and other colleagues hold the copyright of them. However, in order to provide more information to the readers we added the website of the course (removed for Peer Review), which would be available to the readers if the paper is published (page line 313).
Round 2
Reviewer 1 Report
The article has been improved in relation to the provided feedback.
Reviewer 2 Report
The author(s) have made changes to the paper according to the feedback. This has improved the overall quality of the paper.